# Observation of a backward sliding motion for rollers on surfaces in viscoelastic fluid

Chengyu He[1,4], Yateng Qiao[2,4], Yu Cao[1], Boqi Liu[1], Zijie Qu ®[2] ✉, Gaojin Li ®[3] ✉ & Xin Cao ®[1] ✉

When wheels roll on the ground, they move forward via the contact friction. This propulsion mechanism is also frequently used to create artificial micro-swimmers in viscous liquid environment. In such cases hydrodynamic lubrication at the contact point greatly reduces the driving forces, nevertheless forward motion can be still remarkable under rapid rotation. Interestingly, here we demonstrate that when rolling in viscoelastic fluids, a roller can slide backwards even though its rotational direction suggests forward motion. These fluids exhibit along sheared flow lines a non-zero elastic tension due to the stretch of the viscoelastic media. When the roller rolls on a surface within such fluid, a rear-front flow-field asymmetry develops, which leads to a net backward viscoelastic stress that forces the roller to slide backwards. In addition, the viscoelastic flow field results in an effective attraction between the roller and the surface. This allows the assembling of a microscale gearing system which transmit motion from a rotating colloid to a much larger object. Our findings opens up new directions for the fabrication of micro scale actuation systems relevant for active matter design and cargo delivery in complex liquid environment.

Wheeled transportation may have already appeared in stone age when ancient men used rolling logs to effectively transport their cargo[1,2]. This idea is also frequently adopted in recent fabrication of micro-scale active particles or swimmers[3–6], where directed motion in a viscous liquid environment is achieved by rolling on surfaces under applied magnetic[7–10], electric[11–13], or optical field[14–16]. Such micro rollers not only play important roles in understanding collective behaviors of active matter systems[17–20], but also bear great potential for medical applications such as noninvasive surgery and targeted drug delivery[21–24]. In general, when a roller of radius $\vec{r}$ is rotated at speed $\vec{\omega}$, its translational velocity $\vec{v} = k\vec{r} \times \vec{\omega}$, where the coefficient $k$ describes the effectiveness of the rotation-to-translation conversion. Whereas $k = 1$ is easily achieved in dry conditions with the help of gears and rubber tires, this coefficient can be quite small for micro rollers in a liquid environment due to the strong hydrodynamic lubrication between the roller and the underlying surface[25,26]. This has led to the fabrication of various non-

spherical micro rollers rolling on patterned surfaces, where contact slip can be reduced[27,28].

While the behaviors of rollers have been intensively studied in Newtonian fluids, little is known regarding their behaviors in viscoelastic fluids. Such fluids, like many biofluids and polymer solutions, can behave as both liquids and solids with distinct flow-field patterns and stress relaxation mechanisms[29,30]. This often leads to interesting dynamic processes when particulate matter moves in them, such as the self-propulsion of snowman-shaped dumbbells[31], the levitation of spinning objects near a surface[32], the memory-induced oscillations of driven colloids[33], the circular motion of fast self-propelling colloids[34], Magnus effects[35], and more[36,37]. Here, we report the experimental observation of an interesting backward sliding where the rotation-to-translation conversion coefficient $k < 0$ when rollers roll on surfaces in viscoelastic fluids. This observation is robust across different surface roughness, in different viscoelastic fluids, and for different-sized and

[1]School of Physics and Astronomy, Shanghai Jiao Tong University, Shanghai, China. [2]Global College, Shanghai Jiao Tong University, Shanghai, China. [3]State Key Laboratory of Ocean Engineering, School of Ocean and Civil Engineering, Shanghai Jiao Tong University, Shanghai, PR China. [4]These authors contributed equally: Chengyu He, Yateng Qiao. ✉e-mail: zijie.qu@sjtu.edu.cn; gaojinli@sjtu.edu.cn; xin.cao@sjtu.edu.cn

shaped rollers from the micrometer scale to the millimeter scale. Via a computational fluid dynamics simulation, we show that the viscoelastic media surrounding the rotating sphere become strongly stretched due to the strong shear flow. Since there are more streamlines at the back (windward side) of the sphere compared with those in the front (leeward), this results in a net viscoelastic stress that pulls on the sphere from the back. This finally leads to the observed backward sliding if the backward force becomes greater than the forward friction force. In addition to a backward force, the stretched viscoelastic media also generate a normal stress component that acts as an effective attraction between the roller and the surface. This allows the reversible assembling of magnetically controlled micro rotors with larger objects to form a simple gearing system, which bears potential for active matter design and selective cargo delivery in complex liquid environments.

## Results

We prepare rollers at both the micrometer scale and the millimeter scale in our experiments. For micro rollers, we use superparamagnetic colloidal spheres with diameter $D = 2r = 2.8, 4.5, 20, 30\ \mu m$, which are suspended in a viscoelastic fluid contained in a thin sample cell (see Fig. 1a). Due to gravity, the spheres slowly sediment towards the bottom surface of the cell, where their motion is imaged with a video camera via a 40× microscope objective. For millimeter rollers, we use $D = 1.0, 2.0, 4.0, 8.0\ mm$ neodymium sphere magnets, which are immersed in a viscoelastic fluid contained in a petri dish. The motion of these spheres is filmed via a cellphone from different angles. The viscoelastic fluid is, unless otherwise stated, an aqueous polyacrylamide

(PAAM) solution with PAAM molecular weight 18 MDa and various mass concentration $c$. To set the spheres into rotational motion (i.e., spinning), a rotating magnetic field $\mathbf{H}(t)$ is created within the entire sample by using three perpendicular pairs of coils (see Fig. 1a). The spheres rotate along the y-axis when $H_x = H \cos \omega_H t$, $H_z = -H \sin \omega_H t$ and $H_y = 0$ (here $H = |\mathbf{H}|$ is the magnetic field strength, and $\omega_H$ is the angular velocity of $\mathbf{H}$). Rotational motion along different axes can be similarly created. Since the micrometer spheres have negligible residual magnetization, they perform smooth rotation with rotational velocity $\omega \propto H^2$ at sufficiently large $\omega_H$. While neodymium sphere magnets (i.e., millimeter rollers) perform smooth rotation with $\omega = \omega_H$ at sufficiently large $H$. Smaller colloidal spheres ($D = 2.8, 4.5\ \mu m$) were coated with a layer of 50 nm antiferromagnetic chromium on one hemisphere, which leads to a periodic variation of the imaging brightness (Supplementary Fig. 1), which allows us to track the rotation of these spheres. Larger spheres' rotation is tracked via the small features on the surfaces of the spheres. More experimental details regarding sample preparation and rotation mechanism are provided in the "Methods" section.

We firstly rotate the $D = 4.5\ \mu m$ colloidal spheres along the y-axis (i.e., rolling in the x-direction) on a flat surface in water. As expected, the spheres move forward (i.e., in positive x-direction) as shown in Fig. 1b (top). Interestingly, when we rotate these spheres in an aqueous PAAM solution (i.e., viscoelastic fluid) with PAAM mass concentration $c = 0.1\ g/L$, the colloid slides backward (i.e., in the negative x-direction) as shown in Fig. 1b (bottom). Note that we did not observe such backward sliding for rollers in pure viscous fluid (water glycerol mixture with mass ratio 4:6) with similar viscosity compared with our

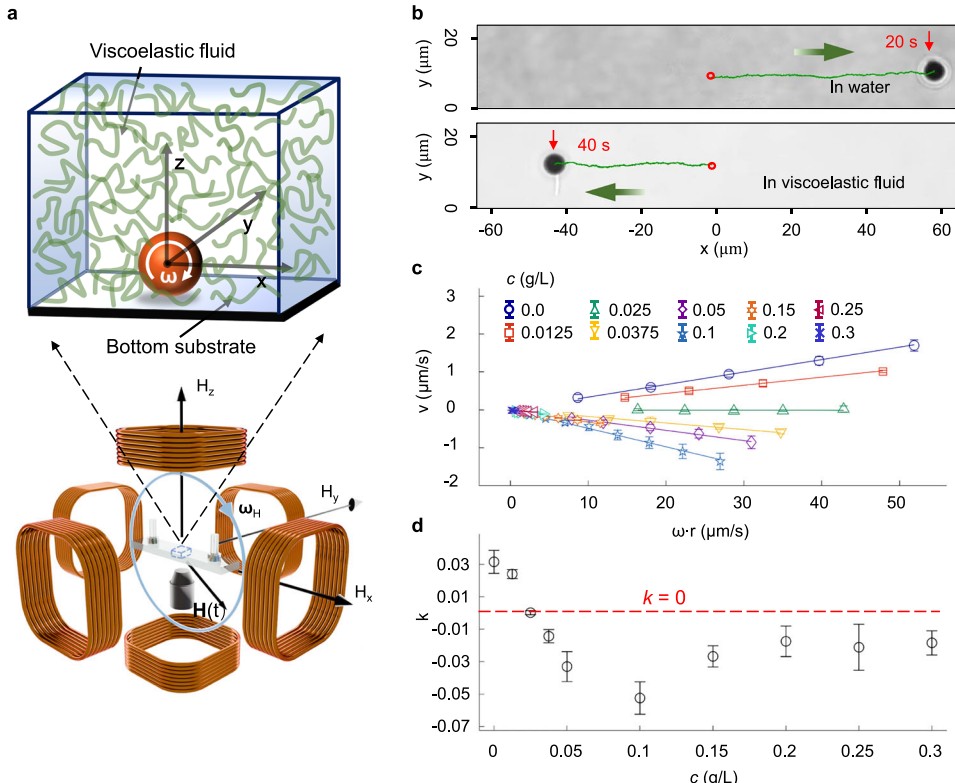

**Fig. 1 | Experimental observation of the backward sliding for micro rollers.**
**a** Sketch of our experimental setup for the motion control and observation of micrometer rollers in a viscoelastic-fluid environment. The roller sediments at the bottom surface of a thin cuvette with an inner space of 20 mm × 10 mm × 0.2 mm and filled with viscoelastic fluid. The rotating magnetic field $\mathbf{H}(t)$ caused the sphere to roll on the surface. **b** Trajectories (green) of the $D = 4.5\ \mu m$ spheres (shown in black) rolling in water (top) with $\omega = 44.88\ rad/s$ and in $c = 0.1\ g/L$ PAAM solution (bottom) with $\omega = 8.62\ rad/s$. The trajectories start at $t = 0$ and $x = 0$ and end at the respective time and position as indicated. **c** Data points are the translational velocity $v$ as a function of the circular velocity $\omega r$ of the $D = 4.5\ \mu m$ roller in PAAM solutions of different $c$ up to $c = 0.3\ g/L$. Lines are corresponding linear fittings. Note that at higher $c$ the data points are localized around the origin because the fluid becomes much more viscous for the micro rollers to move or rotate. **d** The measured $k$, i.e., slopes in (**c**), as a function of $c$. All error bars represent standard deviations of five independent measurements. Source data are provided as a Source Data file.

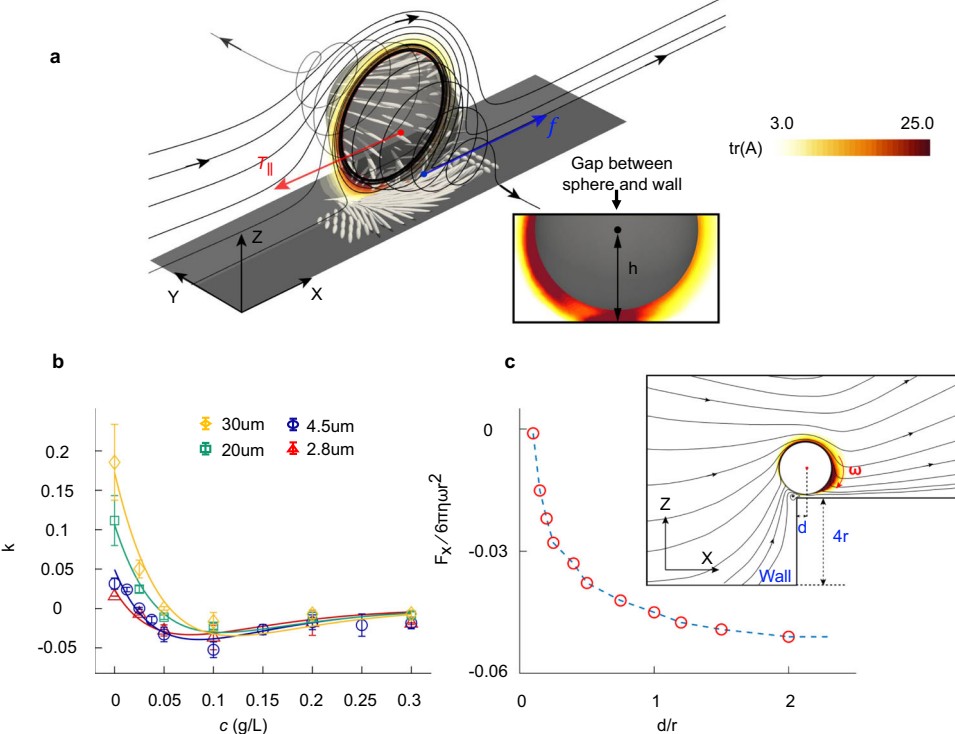

**Fig. 2 | Understanding the backward sliding in a viscoelastic fluid. a** Simulated flow field (lines) for a sphere rotating ($\omega = 1.27$ rad/s) along the y-axis and moving backward ($v = -0.044$ μm/s in the x-direction) on top of a flat surface in a viscoelastic fluid. The sphere radius is $r = 2.25$ μm. Its center of mass is fixed at $h = 1.15r$ away from the bottom surface. $\omega$ and $v$ are chosen such that the sphere is force-free in the x-direction. The liquid domain reaches 60D away from the center of the sphere in all directions except the bottom (no-slip boundary). To simulate the backward motion, the inlet and outlet boundaries are used in the negative and positive x-direction, respectively, with velocity $-v$, such that the sphere is moving with $v$ relative to the liquid domain. The far-field boundary is used in y-directions and at the top. The polymer stretch around the roller is revealed by the axial ratio of the ellipsoids as well as the colormap of the trace tr(A) of the conformation matrix A, see details in Supplementary Note 3. Inset: The flow field and the colormap of tr(A) at the equatorial plane near the gap between the roller and the surface. **b** Data points are experimentally measured $k$ as a function of $c$ for different-sized magnetic

micro rollers ($D = 2r = 2.8, 4.5, 20, 30$ μm). Lines are corresponding fittings to Eq. (2) with fitted $\alpha = 6.25 \times 10^{-5}, 1.36 \times 10^{-4}, 6.73 \times 10^{-4}, 1.40 \times 10^{-3}$ L/s, respectively. The parameter $\gamma_s = 8.38 \times 10^{-7}, 2.91 \times 10^{-6}, 2.97 \times 10^{-5}, 7.81 \times 10^{-5}$ g/s is obtained by the data at $c = 0$ (i.e., in water). Note the onset of backward sliding (i.e., where $k$ reaches 0) increased from $c \sim 0.02$ g/L to $c \sim 0.05$ g/L when roller size increased from 2.8 μm to 30 μm. Error bars are standard deviations of five independent measurements. **c** Data points are the numerically calculated force $F_X$ along the x-direction when a sphere is rotating with $\omega = 1.27$ rad/s at fixed $h = 1.15r$ and fixed distance d to the cliff. Lines are guides for the eye. Inset: Simulated flow field (lines) when a sphere is rotating along the y-axis near a cliff in viscoelastic fluid at fixed $h = 1.15r$ and $d = 0.2r$. The computational geometry is the same as that in (**a**) except that here $v = 0$ and a stair-like surface is used at the bottom with a step height equals $4r$. The colored region represents the polymer stretch with the same color bar as in (**a**). Source data are provided as a Source Data file.

viscoelastic fluid. Further experiments in two other kinds of viscoelastic fluids (a micellar solution composed of 5 mM equimolar cetylpyridinium chloride monohydrate and sodium salicylate in water, and an egg white liquid) revealed similar backward sliding. See Supplementary Movie 1 for a comparison of the rolling behaviors in the mentioned fluids. In addition, backward sliding is also observed with non-spherical rollers, with rollers on rough surfaces, as well as rollers of very different sizes from micrometer scale to millimeter scale, see Supplementary Movie 2. These findings demonstrate that the backward sliding phenomenon is a generic feature for rollers in viscoelastic fluids. For systematic analysis, below we stick our experiments to spherical rollers in PAAM solution on flat surfaces.

For the $D = 4.5$ μm rollers in fluids of various PAAM concentration $c$, the measured sliding velocity $v$ depends linearly on their rotational speed $\omega$ as shown in Fig. 1c. Note the negative slopes that indicate the backward sliding. The linearity in Fig. 1c is in accord with the fact that the PAAM concentration is in the dilute and semi-dilute range ($c \leq$ 0.3 g/L) here, so that entanglements between polymer coils are weak[38–40]. From the slopes in Fig. 1c, we obtain the coefficient $k$, which is plotted as a function of $c$ as shown in Fig. 1d. At $c = 0$ (i.e., in water), $k$ is positive but much smaller than unity as a result of the hydrodynamic lubrication at the contacting point[25]. Upon increasing $c$, $k$ decreases to

zero at around $c = 0.025$ g/L. At this point, the rollers' translational displacement becomes negligible even though they are continuously rotating, see Supplementary Movie 3. $k$ reaches a minimum around $c = 0.1$ g/L and becomes almost constant (remains negative) at larger $c$.

To understand the origin of the negative coefficient $k$, we numerically calculated the viscoelastic fluid flow around a rolling sphere above a surface using the Giesekus model. This model treats the viscoelastic fluid as a combination of a viscous solvent and an elastic polymer network, see Supplementary Note 1 for details. We chose the Giesekus model over the classic Oldroyd-B model due to its improved numerical stability in simulations, particularly given the narrow gap between the rotating sphere and the underlying surface. Nevertheless, both models capture the essential features of the observed backward sliding behavior, as discussed in Supplementary Note 2. Figure 2a shows the simulated flow field when a sphere is rotating along the y-axis with a fixed height $h = 1.15r$ between the sphere center and the surface. As expected, the fluid flows from the negative to the positive x-direction via the top of the sphere. Interestingly, we also observe streamlines spiraling towards both directions of the y-axis, a feature confirmed by our experiments (see Supplementary Movie 4). These perpendicular flow lines provide evidence of elastic tension, known as the normal stresses, in shear flows, which is

responsible for the well-known Weissenberg effect (or rod-climbing effect) where viscoelastic fluid climbs up a partially submerged rotating rod[41]. The spatial distribution of the elastic stress is revealed by the stretched polymers (represented by the ellipsoids) in Fig. 2a. Strong elongation of polymers exists around the sphere, which generates large elastic stress and pulls the sphere from both sides of the sphere.

From Fig. 2a, it is clear that there are more streamlines at the windward side (i.e., in the negative x-direction) of the roller than on the leeward side. This suggests that the elastic tension is larger at the back of the sphere compared with that in the front, considering that shear flow in a viscoelastic fluid stretches the polymers within and generates elastic tension. The larger elastic tension at the back of the sphere is also revealed in Fig. 2a by the colormap of the trace of the conformation matrix at the equatorial plane, which provides a measure of the elastic stress distribution[42,43]. This gives rise to a net backward elastic force $T_{\parallel}$ as illustrated in Fig. 2a. Given the no-slip boundary condition, the fluid at the bottom of the sphere moves at a velocity of $\omega r - v$ relative to the underlying surface. This results in a surface-induced lubrication friction $f = \gamma_s(\omega r - v)$, where $\gamma_s$ is the surface lubrication friction coefficient. Following the Stokes law, the sliding velocity v of the sphere satisfies

$$f - T_{\parallel} = \gamma v \tag{1}$$

where $\gamma$ is the drag coefficient of the sphere moving in the bulk fluid. Equation (1) means the backward sliding occurs when $T_{\parallel}$ exceeds the forward frictional force $f$. This is validated in Supplementary Note 4, where we provide a force analysis based on the Giesekus model. Considering that the polymer concentration used here lies in the dilute and semi-dilute range, we assume a linear response with $T_{\parallel} = \alpha c \omega r$. This gives:

$$k = \frac{v}{\omega r} = \frac{\gamma_s - \alpha c}{\gamma_s + 6\pi r \eta_0 \exp(\beta c)} \tag{2}$$

Here, we have used $\gamma = 6\pi r \eta$ and the experimentally measured viscosity $\eta = \eta_0 \exp(\beta c)$ for the dilute and semi-dilute PAAM solution, where $\eta_0 = 0.0013$ Pa·s the viscosity at $c = 0$ and the fitted $\beta = 15.8$ L/g (see Supplementary Fig. 1). Equation (2) agrees well with the experimentally measured $k$ as a function of $c$ for different-sized micro rollers, see Fig. 2b. This confirms that the backward sliding is a result of a stronger elastic force $T_{\parallel}$ compared with the surface-induced frictional force $f$. It is worth noting that the onset of backward sliding motion occurs at higher polymer concentration for larger-sized rollers (see Fig. 2b). This is because larger rollers have greater friction with the underlying surface. Therefore, Eq. (2) may no longer be valid for sufficiently large rollers, since the required $c$ for the observation of backward sliding motion might become so large (e.g., greater than 0.3 g/L) that linear response is no longer valid[44–46]. This is clearly revealed below for millimeter-sized rollers.

As further evidence that the asymmetric flow field caused the backward motion, in Fig. 2c, our numerical results show that the total force $F_x = f - T_{\parallel}$ in the x-direction rapidly reduces to zero as the rolling sphere approaches a cliff at its back. This suggests that the backward motion would stop, and the roller would not roll off the edge when sliding backward from a distant position towards the cliff. This is indeed confirmed in our experiments as shown in Supplementary Movie 5, for both micrometer-scale and millimeter-scale rollers. Such behavior is also evident in the flow field as shown in the inset of Fig. 2c. Since the streamlines can now enter from below the cliff, this reduces the incoming flow from the negative x-direction and therefore reduces the backward force $T_{\parallel}$.

In contrast to the micrometer rollers, where the backward sliding is observed even in dilute and semi-dilute polymer solutions, for millimeter rollers this is observed only at sufficiently high polymer concentration and large angular velocity $\omega$. This is seen in Fig. 3a, b, where the $\omega$-dependent coefficient $k$ is revealed for the $D = 1$ mm and 4 mm

rollers, respectively, in viscoelastic fluids with varying polymer concentration $c > 0.5$ g/L. Note that $k$ is independent of $\omega$ for micro rollers at small $c$ due to the linear elastic response there. The $\omega$-dependent $k$ in Fig. 3a, b suggests a highly non-linear elastic response for millimeter rollers at large $c$. Interestingly, when we plot these $k$ values as a function of the Weissenberg number Wi $= \omega\tau$ (Wi measures the ratio of elastic force to viscous force in viscoelastic fluids under shear[47]) as shown in Fig. 3c, the data for different $c$ and $\omega$ nicely collapse to the curve

$$k = \frac{k_0 \cdot \text{Wi}_c^2 + k_\infty \cdot \text{Wi}^2}{\text{Wi}_c^2 + \text{Wi}^2} \tag{3}$$

Here, $\tau$ is the stress relaxation time of the viscoelastic fluid which depends on $c$ (see Supplementary Fig. 2), $k_0$ the value of $k$ at low-Wi limit while $k_\infty$ the value of $k$ at high-Wi limit. Note that Eq. (3) can be easily derived from Eq. (1) by taking $T_{\parallel} = C_t \text{Wi}_c^2 v + C_r \text{Wi}^2 \omega r$ with constants $C_t$ and $C_r$. This again confirms that the backward sliding is a result of a stronger backward elastic tension compared with the forward friction force. The parameter $\text{Wi}_c$ in Eq. (3) is a characteristic Weissenberg number required for the backward elastic force to become comparable to the forward friction force. As larger spheres have greater friction with the underlying surface, the corresponding $\text{Wi}_c$ is larger. This is clearly seen in Fig. 3c. We also numerically calculated $k$ as a function of Wi for our Giesekus model for spheres of different distance from the surface, see Fig. 3d. Even though the numerical results have a much smaller $k_0$ due to the large sphere-to-surface gap (see Supplementary Fig. 3 and Movie 6), they capture the essence of Eq. (3) which suggests that $k$ would approach a negative $k_\infty$ at large Wi. It is worth mentioning that the PAAM solutions used for the millimeter rollers exhibit shear thinning behavior due to their large PAAM concentrations $c > 0.3$ g/L (see Supplementary Fig. 4). This will reduce the forward friction force $f$ and therefore facilitate the backward sliding motion. However, the main driving force of the observed backward sliding motion is still the backward viscoelastic force $T_{\parallel}$. This is shown by the agreement of our experimental data to Eq. (3) and also supported by Fig. 2c and Supplementary Movie 5.

Finally, the flow field, as illustrated in Fig. 2a, also leads to an effective attraction between the rolling sphere and the nearby surface. This is confirmed in our experiments where micrometer rollers can roll at the ceiling of a sample ceil and at the vertical side walls of a square pit, see Supplementary Movie 7 and Supplementary Fig. 5. These observations allow us to design a micro-scale gearing system as shown in Fig. 4a. When a magnetic sphere is rotating along z-axis and is located near a larger non-magnetic sphere in viscoelastic fluid, the magnetic sphere will become attracted to the non-magnetic sphere and form an asymmetric dumbbell. Due to the contact friction between the two spheres, rotational motion will then be transmitted (see details in Supplementary Fig. 6) from the magnetic sphere to the large non-magnetic one, causing the later to perform circular motion in the opposite direction as shown in Fig. 4a. See also Supplementary Movie 8 where motion transmission is achieved for magnetic spheres to actuate non-magnetic ones at volume ratios up to 1:154. The radius $R$ of the above-mentioned circular motion depends on the rotation speed $\omega$ of the magnetic sphere, as shown in Fig. 4b. By continuously decreasing the rotation speed of the magnetic sphere, the trajectory of the non-magnetic sphere forms a fascinating spiral pattern, as shown in Fig. 4b inset. Note that the tangential velocity in the circular motion is always perpendicular to the symmetry axis of the dumbbell (see e.g., Fig. 4a, b). This allows us to deliver the non-magnetic sphere to any position in the x–z plane via a programmable control of the magnetic sphere's rotation and rolling, which is demonstrated in Fig. 4c and Supplementary Movie 9.

## Discussion

In summary, the viscoelastic-fluid-flow-induced backward sliding behavior as observed in our experiments is not only of fundamental

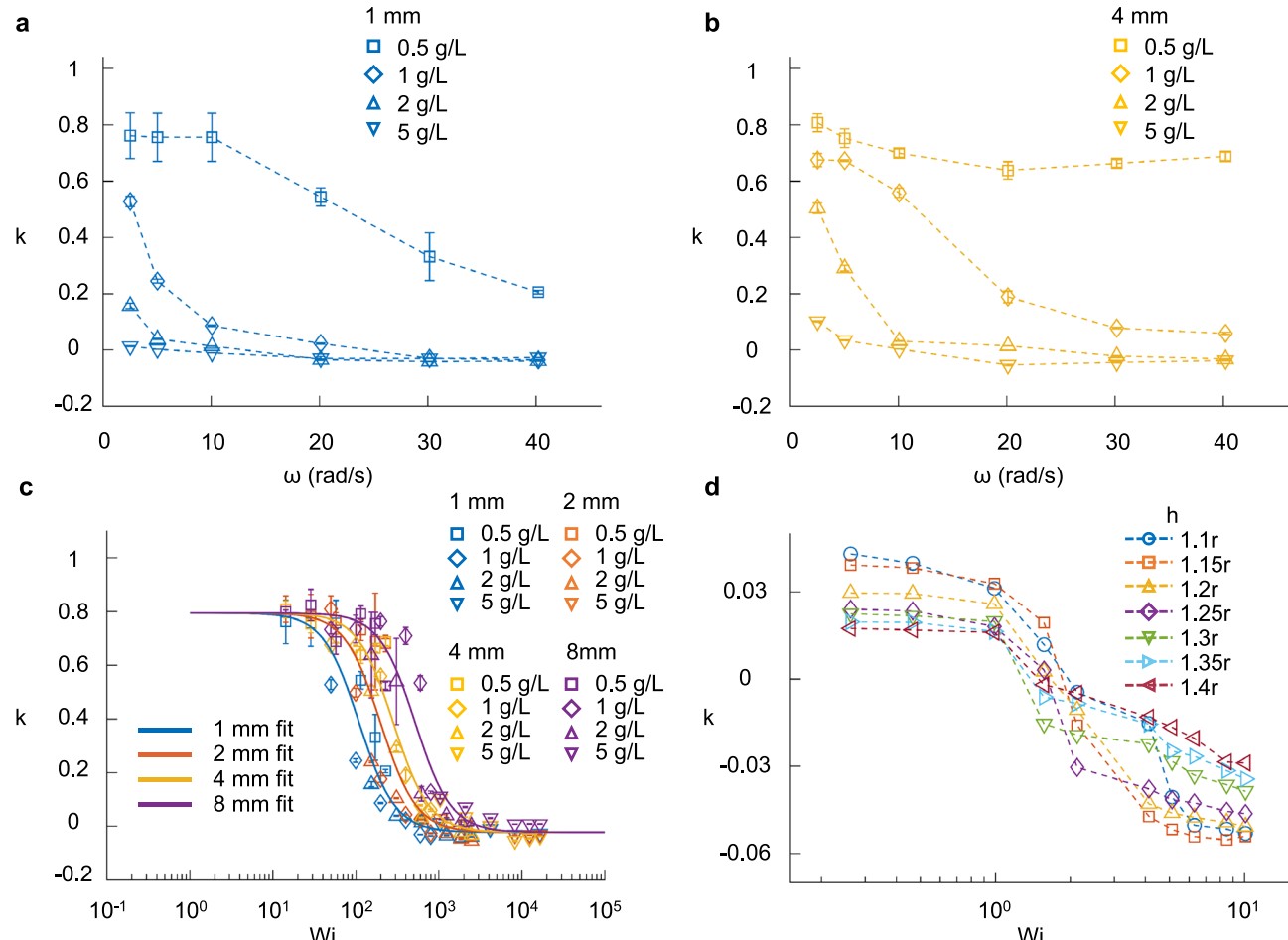

**Fig. 3 | Backsliding behaviors for millimeter rollers. a, b** Data points are experimentally measured $k$ as a function of $\omega$ at different $c$ for 1 mm and 4 mm millimeter rollers, respectively. Lines are guides for the eye. For the 1 mm roller, backward sliding is observed at large $\omega$ for the curves with $c \geq 1\,g/L$. While for the 4 mm roller, backward sliding is observed at large $\omega$ with $c \geq 2\,g/L$. **c** The $k$ data plotted as a function of Wi for different-sized millimeter rollers in viscoelastic fluids of different $c$. Lines are fitted to Eq. (3) with fitted $\mathrm{Wi_c} = 109.5, 198.5, 283.9, 519.1$, respectively, at fixed $k_0 = 0.7943$ and $k_\infty = -0.0217$. **d** Data points are the $k - Wi$ relation obtained in numerical simulations of different h. Lines are guides for the eye. The results are obtained via the force-free condition in the x-direction. All error bars represent standard deviations obtained from five independent measurements. Source data are provided as a Source Data file.

interest to scientists in fields of microrheology and active matter, but also bears potential for developing targeted cargo delivery methodology inside living systems where viscoelastic fluids abounds[48]. In addition to the actuation via rotating magnetic field as used in our experiments, colloidal rollers are frequently actuated by electric field[11–13,49], optical field[14–16,50,51], and in acoustofluidics[52]. This allows a more versatile design of functional microswimmers in combination with the backward sliding behavior observed in our experiments. Finally, our results indicate that the viscoelastic flow field near a rotating object can become strongly influenced in the presence of a wall or other objects. This may lead to interesting collective behaviors when multiple or large numbers of rotors are in the vicinity of one another in viscoelastic fluids.

## Methods

### Preparation of viscoelastic fluids
We prepare three different kinds of viscoelastic fluids. (1) Aqueous solutions of polyacrylamide (PAAM) with a molecular weight of 18 MDa and mass concentrations of 0.0125, 0.025, 0.0375, 0.05, 0.1, 0.15, 0.2, 0.25, 0.3, 0.5, 1, 2, 5, 10, and 20 g/L, respectively, where concentrations ≤ 0.3 g/L are used for microscopic experiments, and concentrations ≥ 0.5 g/L are used for millimeter-scale experiments. (2) An aqueous micellar solution composed of equimolar cetylpyridinium chloride (CPyCl) and sodium salicylate (NaSal) with a molar density of 5.0 mM. (3)

A pure egg white liquid (bio-viscoelastic fluid) prepared by separating fresh eggs from their yolks. The polymer solutions and micellar solutions are kept in glass bottles on top of a magnetic stirrer. This ensures that the polymers and micelles are uniformly distributed within the solution three days after preparation. These solutions are then used within one month before their viscoelastic properties significantly change. While for the egg white fluid, they are prepared upon use.

### Preparation of Janus colloidal spheres
To prepare Janus colloidal spheres, we first prepare monolayers of colloidal spheres (Dynabeads M-450 or M-270 with a diameter of 4.5 and 2.8 μm, respectively, or silica beads with a diameter of 15 or 10.6 μm) on a glass slide via droplet evaporation. A layer of 50-nm-thick chromium is then deposited on the Dynabeads monolayers using electron beam evaporation (DZS-500, Shenyang Scientific Instrument Co., Ltd.). While a layer of 100-nm-thick carbon is deposited on the silica monolayers using magnetron sputtering (SD-650MH, Boyuan Micro-Nano Technology Co., Ltd.). The slide is cleaned with isopropanol and deionized water, and finally sonicated in deionized water so that we can collect the Janus particles.

### Colloidal sample preparation
To prepare a colloidal sample, we disperse the colloidal spheres into 100 microliters of viscoelastic fluid at a number density of ~10⁸ per

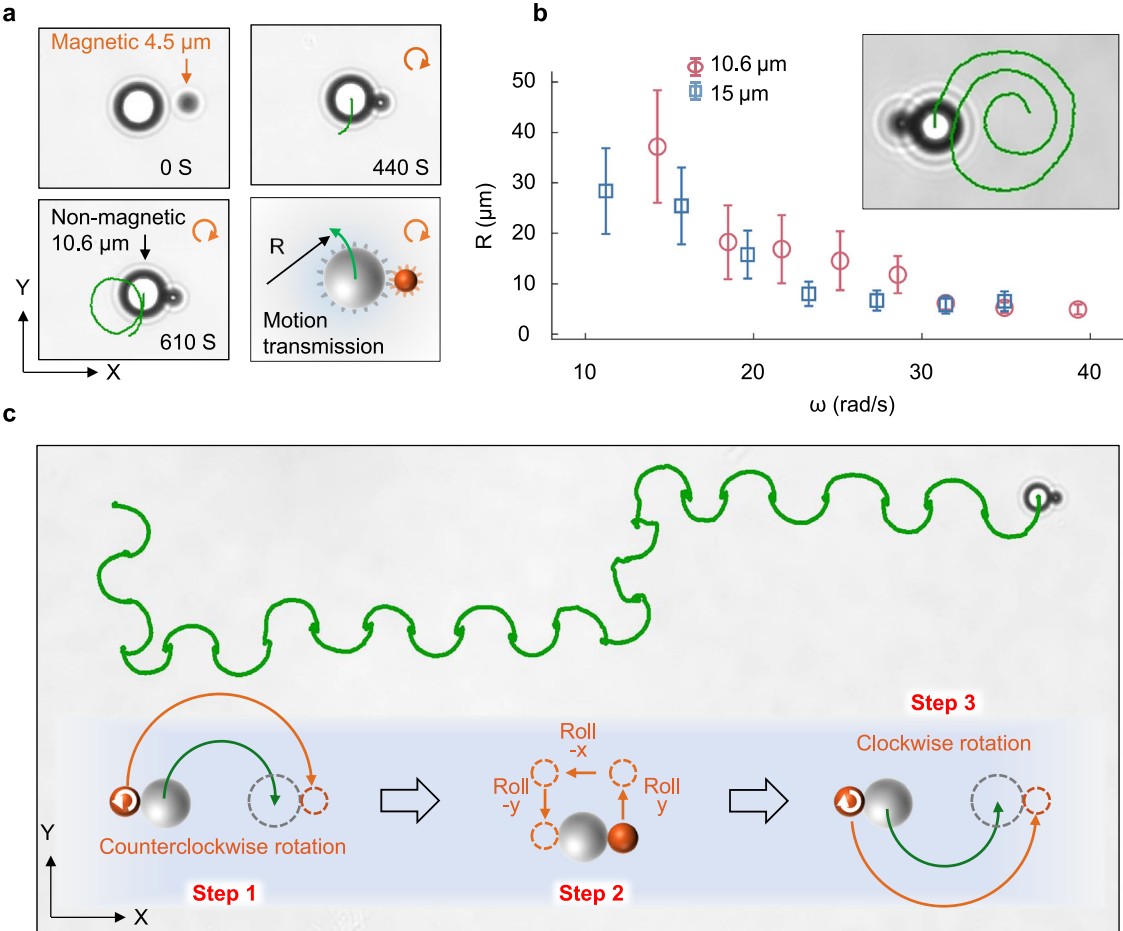

**Fig. 4 | Motion transmission. a** Snapshots ($t$ = 0 s, 440 s, 610 s) showing the formation of an asymmetric dumbbell and the subsequent center-of-mass trajectory (green lines) of the non-magnetic sphere ($D$ = 10.6 μm) due to the rotation ($\omega$ = 14.59 rad/s) of the magnetic one ($D$ = 4.5 μm). The lower-right panel is a schematic of the gearing system. An in-plane rotating magnetic field ($H_x$ = cos $\omega_H t$, $H_y$ = sin $\omega_H t$, and $H_z$ = 0 with $H$ = 2754.18 A/m and $\omega_H$ = 200 πs$^{-1}$) is applied here to achieve the rotation of the magnetic sphere along the z-axis. **b** The measured radius R for the circular trajectories of two different-sized non-magnetic spheres, respectively, as a function of the angular velocity $\omega$ of the magnetic sphere ($D$ = 4.5 μm). Error bars are standard deviations of five independent measurements. Inset: A spiral trajectory of the 10.6 μm sphere formed via continuously changing $\omega$

of the magnetic sphere from 17.72 rad/s to 9.15 rad/s over a period of 460 s. **c** Trajectory (green line) of a non-magnetic sphere obtained via a programmable control as demonstrated with the following steps. Step 1: The magnetic sphere is positioned to the left of the non-magnetic one (via rolling) and then rotated counterclockwise until the non-magnetic sphere is moved by half a circle (shown in green). Step 2: The magnetic sphere is repositioned to the left of the non-magnetic sphere via rolling in y, −x, and −y directions, respectively. Step 3: The magnetic sphere is rotated clockwise until the non-magnetic sphere is moved by half a circle. Via repeating Steps 1–3, the large sphere can be carried along the x-direction. Similar steps can be taken to carry the large sphere to any direction in the x–y plane. Source data are provided as a Source Data file.

liter, which corresponds to one or two spheres within the field of view (180 × 280 μm$^2$) of the microscope. The colloidal suspension is then injected into a glass cuvette with inner dimensions of approximately 20 × 10 × 0.2 mm$^3$, or into a homemade thin sample cell by placing a cover slide on top of a substrate slide spaced by parafilm in between and sealed with epoxy glue. The homemade sample cell is used when the substrate needs to be structured, while the cuvette is used otherwise. The prepared sample is then transferred to the stage of an inverted microscope (NIB900, Ningbo Yongxin Optics Co., Ltd, China), where a rotating magnetic field can be generated using the magnetic coils and the temperature is maintained at 25 ± 1 °C using a water bath.

### Structured substrate preparation

We prepare two kinds of structured substrates via photolithography. (1) A rough substrate composed of periodically arranged shallow holes (0.17 μm deep and 2 μm diameter) with lattice spacing (triangle lattice) of 3.2 μm. (2) A substrate that contains square pits with a side length of 100 μm and a depth of 15 μm. The walls of the square pits act as cliffs mentioned in Fig. 2c. For the periodically structured substrate, we first spin-coated the photoresist (SU8 2000.1) on a glass slide at 3000 rpm

for 30 s. After soft baking (95 °C, 1 min), the slide is then exposed with 720 mJ/cm$^2$ under a 365 nm UV LED lamp (YUNHOE UVDL-L120W120) through a photomask that contains the periodic pattern. Afterwards, the slide is post-baked (95 °C, 1.5 min), before it is developed in SU8 developer (40 s) and washed with isopropanol (5 s). The resulting periodic structure showed an average peak-to-valley distance of 169 nm by AFM (Cypher S, Oxford Instruments). To make the square pit structure, the procedure is similar by using SU8 2015 and a photomask with a corresponding square structure, with longer soft baking (3 min) and post baking (8 min), and we added a final hard bake (150 °C, 6 min) to eliminate the residual stress after washing with isopropanol.

### Generation of a rotating magnetic field

We use three pairs of mutually perpendicular coils to generate magnetic fields in our colloidal sample. Each pair of coils is connected to a power amplifier (Aigtek ATA-309) controlled by a sinusoidal waveform generator (Rigol DG1022Z). This allows us to fully control each of the magnetic components $H_x(t) = H_1 \cos(\omega_1 t + \phi_1)$, $H_y(t) = H_2 \cos(\omega_2 t + \phi_2)$, and $H_z(t) = H_3 \cos(\omega_3 t + \phi_3)$. For example, setting $H_1 = H_2 = H$, $H_3 = 0$, $\omega_1 = \omega_2 = \omega_H$, and $\phi_1 - \phi_2 = \pi/2$, we generate a rotating magnetic field with

amplitude $H$ and angular velocity $\omega_H$ rotating anti-clockwise in the x–y plane. Similarly, setting $H_1 = H_3 = H$, $H_2 = 0$, $\omega_1 = \omega_3 = \omega_H$, and $\phi_3 - \phi_1 = \pi/2$, we generate a rotating magnetic field with amplitude $H$ and angular velocity $\omega_H$ in the x–z plane.

### Rotation of superparamagnetic colloidal spheres

For superparamagnetic colloidal spheres, at sufficiently large $\omega_H$, the rotating $\mathbf{H}(t)$ induces a phase-lagged magnetization $M(t)$ within the colloidal sphere, which leads to a torque $\Gamma = |\mathbf{M} \times \mathbf{H}| = \gamma_m H^2$ and hence the smooth colloidal rotation[53]. Here, $\gamma_m$ is a parameter that depends on the magnetic susceptibility, the magnetic relaxation time, and the magnetic field frequency. The rotating velocity of the colloidal sphere $\omega = \Gamma/(\pi \eta \sigma^3) \ll \omega_H$, where $\eta$ is the viscosity of the liquid suspension. In our experiments, we set $\omega_H = 20\pi\,\text{rad/s}$, which is already sufficient to realize smooth colloidal rotation. We have determined $\gamma_m$ for the magnetic spheres with a diameter of $D = 4.5\,\mu\text{m}$ in our experiments. This is done by measuring their rotating speed $\omega$ in water with viscosity $\eta = 1.3 \times 10^{-3}\,\text{Pa·s}$ as a function of the applied $H$, see Supplementary Fig. 1. The data fitted nicely to $\omega = k_m H^2$ with fitted $k_m = (7.9 \pm 0.3) \times 10^{-6}\,\text{m}^2/(\text{A}^2\text{s})$. Considering $\gamma_m H^2 = \pi \eta \sigma^3 \omega$ for viscous torque to balance the magnetic torque, this finally gives $\gamma_m = \pi \eta \sigma^3 k_m = 2.94 \times 10^{-6}\,\text{pN·}\mu\text{m}\,\text{A}^{-2}\text{m}^2$.

### Rotation of millimeter neodymium sphere magnets

For neodymium sphere magnets, their large residual magnetization $M_0$ quickly aligns with the external magnetic field $H(t)$. This leads to their rotation with $\omega = \omega_H$ once $H$ is sufficiently large. In our experiments, we set $H = 475\,\text{A/m}$, which is sufficient to ensure that neodymium sphere magnets rotate smoothly with angular velocity $\omega \leq 6.4\,\text{Hz}$.

### Relaxation time measurement

To measure the relaxation time of the dilute and semi-dilute PAAM solutions ($c \leq 0.3\,\text{g/L}$), we first create colloidal trimers in the PAAM solution (formed by three $4.5\,\mu\text{m}$ magnetic spheres when they are close to one another in a rotating magnetic field[35]) and rotate them along the z-axis with fixed angular velocity. We then turn off the magnetic field and measure the angular relaxation of the colloidal trimmer. This gives the relaxation time of the PAAM solution as shown in Supplementary Fig. 2. For higher PAAM concentrations (0.5–5 g/L), we used a rotational rheometer (DHR20, TA Instruments, USA) and performed creep-recovery tests to determine the relaxation time, see Supplementary Fig. 2.

## Data availability

All data generated in this study are provided in the Supplementary Information and Source data files with this paper. Source data are provided with this paper.

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

## Acknowledgements

We acknowledge helpful discussions with Clemens Bechinger, Matthias Krüger, and Hepeng Zhang. X.C. acknowledges funding from the Natural Science Foundation of China (grant no. 12574228) and the start-up fund of Shanghai Jiao Tong University, as well as support from the Yangyang development fund. G.L. acknowledges funding from NSFC (grant nos. 12372264, 12102258) and the Natural Science Foundation of Shanghai (grant no. 23ZR1430800). Z.Q acknowledge fundings from NSFC (grant no. 12202275). Materials fabrication in this work is supported by the Micro-nano Fabrication Platform of the School of Physics and Astronomy at SJTU.

## Author contributions

X.C. conceived the idea and designed the experiments, C.H. performed the experiments and analyzed the experimental data, Y.Q. performed the numerical simulation supervised by G.L. and Z.Q., Y.C. fabricated the patterned substrates and characterized their profile, B.L. fabricated the Janus particles, X.C. and G.L. formulated the theoretical picture, C.H., Y.Q., G.L., and X.C. drafted the manuscript, all authors discussed the results and revised the manuscript.

## Competing interests

The authors declare no competing interests.
