## [Transparent Peer Review file · Nature Communications]

Observation of a backward sliding motion for rollers on surfaces in viscoelastic fluid

Corresponding Author: Professor Xin Cao

Version 0:

Reviewer comments:

Reviewer #1

(Remarks to the Author)
Please see attached PDF

Reviewer #2

(Remarks to the Author)

This manuscript presents an experimental demonstration of a novel propulsion mechanism in viscoelastic fluids. The authors show that when a roller moves through a viscoelastic fluid, it can exhibit backward sliding despite rotating in a direction that typically implies forward motion. This counterintuitive behavior arises from the elastic tension generated along sheared streamlines due to the stretching of the viscoelastic medium. As the roller interacts with the surface, an asymmetric flow field develops between its rear and front, producing a net viscoelastic stress that drives the roller backward.

For me, there are two important contributions of this work:

- **Demonstration of the backward sliding in a viscoelastic fluid.** (Fig. 1 and Fig. 3). This propulsion mechanism opens new possibilities for developing a targeted cargo delivery methodology inside living systems where viscoelastic fluids are omnipresent.
- **Development of a simplified model and viscoelastic numerical simulations** that explain the physical mechanism behind backward motion.

The manuscript is fairly well-written, and the results appear to be sound. I believe this work will be of interest to a broad scientific and engineering audience, including readers of Nature Communications.

However, I have several comments and suggestions that I would like the author to address before I can recommend this work for publication in Nature Communications.

1. Figure 2a: This figure shows the simulated flow field (lines) around a sphere rotating about the y-axis near a surface in a viscoelastic fluid. The authors use this figure to illustrate the proposed mechanism, as discussed in lines 149–152. I find the current explanation unsubstantiated and unconvincing. Rather than relying solely on the simulated flow field, I strongly recommend that the authors include additional visualizations—such as colormaps of the viscoelastic stress tensor components, the trace of the polymeric stress tensor, $\text{tr}(\tau_p)$ ($\text{tr}(\Gamma_p)$), or the trace of the conformation tensor, $\text{tr}(A)$. These quantities would provide a more informative and insightful view of the elastic stress distribution and its role in the observed backward sliding phenomenon. For reference, the authors may consider the approach used in the work of Ardekani and co-workers (<https://doi.org/10.1073/pnas.2211347120>), (<https://doi.org/10.1103/PhysRevFluids.6.033304>), where such visualizations have been effectively employed to elucidate viscoelastic flow mechanisms.

2. Rheology of the PAAm solution: I believe it is very important to include, in the Supplementary Information, the shear and oscillatory rheology of the PAAm solutions at the different concentrations used in this study. This is crucial because PAAm is not only a viscoelastic fluid but also exhibits shear-thinning behavior. Therefore, the authors should discuss how the

interplay between viscoelasticity and shear-thinning behavior influences the observed phenomenon of backward sliding.

3. Figure 1c: This figure is not clear and should be improved. Could the authors explain how the error bars are defined? Furthermore, why do the cyan data points (which I believe correspond to $c = 0.1$) have such large error bars? Please discuss this in the manuscript. In addition, to aid the reader, consider using different symbols in addition to colors to distinguish the data series.

4. The authors appear to have overlooked several relevant prior studies in their literature survey, particularly those addressing the propulsion of viscoelastic fluids under rotating magnetic fields.

- Viscoelastic propulsion of a rotating dumbbell <https://doi.org/10.1007/s10404-019-2275-1>

- Viscoelastic levitation <https://doi.org/10.1017/jfm.2022.418>

5. SI: Before Eq. (S1) that appears in the non-dimensional form: Please provide proper normalization.

6. Supplementary Note 3: $\Gamma p = (\tau/\beta p)(A - l)$ is incorrect. It should be $\Gamma p = (\beta p/Wi)(A-l)$ (see, for example, JFM 2020, <https://doi.org/10.1017/jfm.2020.456>).

The authors should carefully verify the correctness of their normalization procedure.

Minor comments:

1. Lines 60-62: "Since the flow field is stronger at the back of the sphere compared with that in the front, this results in a viscoelastic stress that pulls on the sphere from the back." The first part of the sentence is unclear, and I suggest rewriting it.

2. Lines 63-64: "In addition to a backward stress...". It is unclear what backward stress means.

3. Line 72: Symbol of diameter of sphere: I suggest replacing "sigma" with "D".

4. Line 84: Please define H and $\omega_{\{H\}}$ clearly here. It is hard to understand the difference between $\omega_{\{H\}}$ and $\omega_{\{H\}}$.

5. Figure 1d: To assist the reader, please add a dashed horizontal line (e.g., in red) corresponding to $k=0$.

Reviewer #3

(Remarks to the Author)

The manuscript entitled "Observation of a backward sliding for rollers on surfaces in viscoelastic fluid" reports a counterintuitive phenomenon: rotating spheres ("rollers") translating backwards while rolling along a surface in viscoelastic fluids, an effect absent in Newtonian liquids. The authors combine experiments at different scales and with various viscoelastic media with numerical simulations based on the Giesekus model. They further show that viscoelastic normal stresses can generate an effective attraction to nearby surfaces, enabling the transmission of motion to larger passive spheres in a microscale gearing configuration. The topic falls well within the scope of Nature Communications, the results are interesting and novel, and the work adds to the literature on microrheology and active matter. However, several important issues must be addressed in a comprehensive revision before the manuscript can be considered further.

Comments

Clarity of Figure 1a (experimental schematic).

Figure 1a is too small and visually confusing for a reader encountering the problem for the first time. Because it introduces the setup and defines the physical problem, it should be enlarged and clearly annotated. At first reading, it is not clear whether the roller is fully immersed in the viscoelastic fluid or partly at a free surface; only later in the text does it become apparent that the particles sediment and roll along the bottom wall, fully immersed. This point should be explicitly stated early in the manuscript, and the schematic should include clear labels showing the wall, fluid region, coordinate system, rotation direction, and magnetic field orientation.

Giesekus model parameters and physical consistency.

The parameters used in the Giesekus model require much stronger justification. The mobility parameter $\alpha = 0.2$ is very large and inconsistent with the dilute PAAM solutions studied here, where α values below 0.01 are more realistic. Such a high α imposes strong shear-thinning and large second normal stress differences that are not representative of the experiment, even though this effect is partially suppressed by choosing $\beta = 0.9$. This discrepancy is evident in Fig. 3d, where the simulated coefficient k is underestimated by about one order of magnitude and the transition from positive to negative k occurs at Weissenberg numbers roughly three orders of magnitude smaller (≈ 10 in the simulation versus ≈ 1000 in the experiment). The parameter choice therefore does not capture the experimental regime. Moreover, for smaller and more realistic α values (e.g., 0.01 or less), the predicted normal stresses increase, which would shift the onset of backward sliding to lower Wi , not higher. If backward motion indeed appears at lower Wi in that case—as expected from the Oldroyd-B limit—the normal-stress-driven mechanism proposed here may not be correct. The authors should clarify whether backward sliding occurs for $\alpha = 0.01$ or in the Oldroyd-B model, and if so, whether the transition shifts to lower or higher Wi . This is critical for validating the proposed physical mechanism.

Unclear geometry in Figure 2.

The geometry depicted in Figure 2 is difficult to understand. It is not clear what the "expansion" represents, what the domain

boundaries are, or how the illustrated flow corresponds to the experimental setup in Figure 1a. The figure and caption should explicitly indicate whether the sphere is rotating above a flat wall or near a “cliff,” what the boundary conditions are on each side, and what quantity is being visualized (velocity field, polymer stretch, or stress). A short paragraph in the text explaining the computational geometry and its relation to the experiments would greatly improve readability and credibility of the simulation results.

Version 1:

Reviewer comments:

Reviewer #1

(Remarks to the Author)

The authors addressed all of my previous comments and concerns. I recommend the present form of the revised version for publication in Nature Communications.

Reviewer #2

(Remarks to the Author)

The authors have improved the clarity of the paper and thoroughly addressed all my comments and suggestions. I am happy to recommend this nice work in its current form for publication in Nature Communications.

Reviewer #3

(Remarks to the Author)

The authors have addressed all of my comments. I recommend acceptance of the manuscript.

Reviewer #1 (Remarks to the Author):

The paper investigates the backward sliding motion of magnetic rollers near surfaces in complex fluids. The authors present convincing experiments of the observed dynamics, as well as evidence of hydrodynamic coupling between the rollers and the surface. Numerical simulations complement the observations and provide some explanation for the dynamics, revealing tension force that opposes the natural friction force between the rollers and the surface. After carrying out a force analysis, the authors determined conditions under which the backward sliding motion is likely to occur.

Overall the paper is very well written, the experimental methodology leads to clear and strong evidence of the dynamics, and the theoretical analysis is sound. The results are noteworthy; to my knowledge, reports of backward motions are only now beginning to emerge in the low-Reynolds propulsion literature. [For instance, Della-Giustina et al. (Physics of Fluids, 2023) in the case of an active particle encapsulated in a droplet in a porous medium.] Although using a much different physical set-up compared to previous studies, the authors' findings provide further evidence of counterintuitive dynamical processes taking place in low-Reynolds number flows of complex fluids.

It is my opinion that the findings will expand the scientific understanding of microorganisms' dynamics in biologically-relevant environments. From a practical viewpoint, they also have the potential to guide the design of more efficient drug delivery cargos, thereby serving as the foundation for future studies. Audiences in fluid dynamics and related interdisciplinary fields will benefit from the paper's findings. I highly recommend the manuscript for publication in Nature Communications, and only have the following very minor comments.

Reply: We highly appreciate the reviewer's positive assessment of our work. We also thank the reviewer for mentioning the interesting work of Della-Giustina et al. We have added this in the reference list in our revised manuscript where we mention interesting dynamics of particulate matter in viscoelastic fluid (see ref. 36 of the revised manuscript).

1. The title could benefit from a bit more clarity. I would suggest "Observation of a backward sliding *motion* for rollers on surfaces in viscoelastic fluids".

Reply: Thanks to the suggestion of the reviewer. We fully agree that, including "motion" in the title would increase the clarity. This has been added in our revised manuscript.

2. Line 167: I suggest replacing the text "Eq. (2) fits nicely to" with "Eq. (2) agrees well with".

Reply: Following the suggestion of the reviewer, we have changed the phrase accordingly, see line 222 of the revised manuscript.

3. The authors note that Eq. (2) may not be valid at large polymer concentration. Is there a way to quantify the range of polymer concentration for which Eq. (2) remains a good approximation? In other words, what is considered large? (I presume the answer will depend on the size of the rollers.) A more detailed note on this limitation may be useful.

Reply: The reviewer is right that, the validity of Eq. (2) in describing the backward sliding motion would depend on the size of the rollers.

For larger rollers, their friction with the underlying surface will also become larger. Therefore, to observe backward sliding motion for larger rollers, this will require a larger backward force and hence a larger polymer concentration c according to Eq. (2). When the roller become large enough, the friction become so large that the required c for backward motion would become greater than a so called critical concentration c^* ($c^*=0.224$ g/L for the polymer we have used, see ref. [38-40] in our revised manuscript), above which the polymer coils in the fluid start to become overlapping. In such situation, the linear response as used to derive Eq. (2) become no longer valid, and so does Eq. (2). In general, for micro rollers up to 30 microns, backward motion occurs already at $c = 0.05$ g/L $\ll c^*$, see Fig.2b. Therefore Eq. (2) is well valid, see also our Fig.2b which plotted the experimental data together with Eq. (2) in the range of $c < 0.3$ g/L. In contrast, for the millimeter rollers, the required c for the observation of backward motion reaches 1 g/L $\gg c^*$ as shown in Fig. 3a. Therefore Eq. (2) become no longer valid and one needs to use Eq. (3).

We thank the reviewer for pointing out this question. We have made the following changes to our manuscript to clarify this point.

We have changed "It is worth noting that at large polymer concentration, Eq. (2) may no longer be valid due to the polymer entanglement which often leads to a highly non-linear elastic forces [41 - 43]" (line 170-172 of the old manuscript) to "It is worth noting that, the onset of backward sliding motion occurs at higher polymer concentration for larger sized rollers (see Fig. 2b). This is because larger rollers have greater friction with the underlying surface. Therefore, Eq. (2) may no longer be valid for sufficiently large rollers, since the required c for the observation of backward sliding motion might become so large (e.g. greater than 0.3 g/L) that linear response is no longer valid [44-46]. This is clearly revealed below for millimeter sized rollers." (which appears in line 225-231 of the revised manuscript). We also added necessary descriptions in figure captions (Fig.2b and Fig. 3ab) to help clarify this point.

Reviewer #2 (Remarks to the Author):

This manuscript presents an experimental demonstration of a novel propulsion mechanism in viscoelastic fluids. The authors show that when a roller moves through a viscoelastic fluid, it can exhibit backward sliding despite rotating in a direction that typically implies forward motion. This counterintuitive behavior arises from the elastic tension generated along sheared streamlines due to the stretching of the viscoelastic medium. As the roller interacts with the surface, an asymmetric flow field develops between its rear and front, producing a net viscoelastic stress that drives the roller backward.

For me, there are two important contributions of this work:

- Demonstration of the backward sliding in a viscoelastic fluid. (Fig. 1 and Fig. 3). This propulsion mechanism opens new possibilities for developing a targeted cargo delivery methodology inside living systems where viscoelastic fluids are omnipresent.
- Development of a simplified model and viscoelastic numerical simulations that explain the physical mechanism behind backward motion.

The manuscript is fairly well-written, and the results appear to be sound. I believe this work will be of interest to a broad scientific and engineering audience, including readers of Nature Communications.

Reply: We are very thankful to the reviewer for the positive assessment of our manuscript. Below we answer his/her comments and suggestions point by point.

However, I have several comments and suggestions that I would like the author to address before I can recommend this work for publication in Nature Communications.

1. Figure 2a: This figure shows the simulated flow field (lines) around a sphere rotating about the y-axis near a surface in a viscoelastic fluid. The authors use this figure to illustrate the proposed mechanism, as discussed in lines 149–152.

I find the current explanation unsubstantiated and unconvincing. Rather than relying solely on the simulated flow field, I strongly recommend that the authors include additional visualizations—such as colormaps of the viscoelastic stress tensor components, the trace of the polymeric stress tensor, $\text{tr}(\tau_p)$ ($\text{tr}(\Gamma p)$), or the trace of the conformation tensor, $\text{tr}(A)$. These quantities would provide a more informative and insightful view of the elastic stress distribution and its role in the observed backward sliding phenomenon. For reference, the authors may consider the approach used in the work of Ardekani and co-workers (<https://doi.org/10.1073/pnas.2211347120>), (<https://doi.org/10.1103/PhysRevFluids.6.033304>), where such visualizations have been effectively employed to elucidate viscoelastic flow mechanisms.

Reply: We agree with the reviewer that, including additional visualizations in our Figure

2a will provide more insightful view of the elastic stress distribution and therefore help the readers to understand the backward sliding motion. We have therefore added the colormap of the conformation tensor $\text{tr}(A)$ in the revised version of our manuscript, following the approach used in ref. 42 and ref. 43 of the revised manuscript (see line 205 of the revised manuscript). From the new Figure 2a (in particular its inset), it is clear that $\text{tr}(A)$ is in general larger at the back of the roller compared with that in the front, suggesting a greater elastic force at the back of the roller. To clarify this point, we have also made the following changes to our manuscript.

We have changed "This suggests that the elastic tension is larger at the back of the sphere compared with that in the front" to "This suggests that the elastic tension is larger at the back of the sphere compared with that in the front, considering that shear flow in viscoelastic fluid stretches the polymers within and generates elastic tension. The larger elastic tension at the back of the sphere is also revealed in Fig. 2a by the colormap of the trace of the conformation matrix at the equatorial plane which provide a measure of the elastic stress distribution" (which appears in line 199-205 of the revised manuscript). We have also changed Fig.2a and the corresponding figure caption.

2. Rheology of the PAAm solution: I believe it is very important to include, in the Supplementary Information, the shear and oscillatory rheology of the PAAm solutions at the different concentrations used in this study. This is crucial because PAAm is not only a viscoelastic fluid but also exhibits shear-thinning behavior. Therefore, the authors should discuss how the interplay between viscoelasticity and shear-thinning behavior influences the observed phenomenon of backward sliding.

Reply: Following the suggestion of the reviewer, we have added shear and oscillatory rheology measurements of the PAAM solutions to the supplementary material. In particular, by using a rotational rheometer, we have measured the viscosity as a function of the shear rate, the storage modulus and loss modulus as a function of oscillatory frequency, see Supplementary Fig. 4 in the revised manuscript.

From the new Supplementary Fig. 4, it is clear that, within the range of shear rate considered, the PAAM solution reveals little shear thinning behavior at polymer concentration $c < 0.3$ g/L. This is consistent with our measured linear relation between v and ω (see Fig. 1c) for micrometer rollers rolling in polymer solutions at $c \leq 0.3$ g/L. Therefore, shear thinning has little to do with the observed backward sliding motion for our micro rollers. Such motion for the micro rollers is a direct result of the fluid viscoelasticity, which is described by Eq. (2) of our manuscript.

When the polymer concentration $c \geq 0.5$ g/L, the new Supplementary Fig. 4 shows that the PAAM solutions exhibit clear shear thinning behavior, as pointed out by the

reviewer. This might be relevant for the observed backward sliding motion of our millimeter rollers, where we have used polymer solutions with $c \geq 0.5$ g/L in order to (induce a large enough backward viscoelastic force and) observe the backward sliding motion. While we are not exactly sure about the role of shear thinning, we would expect that it lead to a reduction in the forward friction force f due to the decrease of viscosity when shear thinning happens. According to our Eq. (1), a reduction in f will facilitate the backward sliding motion because it would reduce the required backward viscoelastic force. However, we would not expect that, shear thinning alone (i.e. without the need of a backward viscoelastic force) could lead to the observed backward sliding motion in our experiments. This is supported by the results in Fig. 2c and Supplementary Video 5, which shows that rollers sliding backward toward a cliff in PAAM solution will stop their backward sliding motion and remain on top of the cliff, even though the rollers are still rotating (i.e. shear thinning is still happening). Therefore, we conclude that shear thinning will facilitate backward sliding for the millimeter rollers, but the main driving force for their backward sliding motion is still the viscoelastic force.

Thanks to the comments of the reviewer, we have added the following discussion at line 282 in the revised manuscript.

"It is worth mentioning that, the PAAM solutions used for the millimeter rollers exhibit shear thinning behavior due to their large PAAM concentrations $c > 0.3$ g/L (see Supplementary Fig. 4). This will reduce the forward friction force f and therefore facilitate the backward sliding motion. However, the main driving force of the observed backward sliding motion is still the backward viscoelastic force T_{\parallel} . This is shown by the agreement of our experimental data to Eq. (3) and also supported by Fig. 2c and Supplementary Video 5."

3. Figure 1c: This figure is not clear and should be improved. Could the authors explain how the error bars are defined? Furthermore, why do the cyan data points (which I believe correspond to $c = 0.1$) have such large error bars? Please discuss this in the manuscript. In addition, to aid the reader, consider using different symbols in addition to colors to distinguish the data series.

Reply: The error bars of the data points in Figure 1c are defined as the standard deviation of 5 independent measurements of the corresponding velocity (i.e. from five independent trajectories as shown in Fig. 1b). Thanks to the reviewer for pointing this out, in our revised manuscript we have now added definitions for this and all other error bars in the corresponding figure captions.

Regarding the cyan data points (corresponding to $c = 0.1$ g/L), they have larger error bars because they involve measurements from a very old (a few months old) polymer solution, while all other measurements used freshly prepared PAAM solutions (< 1 months). To be more consistent in our manuscript, we have prepared a fresh PAAM solution and measured the results for $c = 0.1$ g/L again. The result is updated in Figure 1c of the revised manuscript, and other related plots. We also added descriptions

regarding the limited life time of the PAAM solutions to the Method section of the revised manuscript. Comparing the updated figure with the old one, we see that the mean value of the cyan data remains almost the same while the error bars has decreased.

Following the suggestion of the reviewer, in the revised Figure 1c, we have used different symbols in addition to colors to distinguish the data series.

4. The authors appear to have overlooked several relevant prior studies in their literature survey, particularly those addressing the propulsion of viscoelastic fluids under rotating magnetic fields.

- Viscoelastic propulsion of a rotating dumbbell <https://doi.org/10.1007/s10404-019-2275-1>

- Viscoelastic levitation <https://doi.org/10.1017/jfm.2022.418>

Reply: We thank the reviewer for the comments. These references are indeed very relevant to our work. We have now briefly mentioned these works in our introduction and included them in the reference list (see ref. 31 and ref. 32 of the revised manuscript).

5. SI: Before Eq. (S1) that appears in the non-dimensional form: Please provide proper normalization.

Reply: Thanks to the reviewer for pointing this out. We have now added the scaling factors for the non-depersonalization procedure in the Supplementary Information.

6. Supplementary Note 3: $\Gamma_p = (\tau/\beta_p)(A - I)$ is incorrect. It should be $\Gamma_p = (\beta_p/W_i)(A-I)$ (see, for example, JFM 2020, <https://doi.org/10.1017/jfm.2020.456>).

The authors should carefully verify the correctness of their normalization procedure.

Reply: We thank the reviewer for pointing out this mistake. We have now changed the expression to $\Gamma_p = (\beta_p/W_i)(A-I)$. We have also verified the normalization procedure through out our manuscript and corrected an inconsistency regarding the expression of the forces in the Supplementary Information.

Minor comments:

1. Lines 60-62: "Since the flow field is stronger at the back of the sphere compared with that in the front, this results in a viscoelastic stress that pulls on the sphere from the back." The first part of the sentence is unclear, and I suggest rewriting it.

Reply: Following the suggestion of the reviewer, we have now rewritten the first part of the sentence as follows. "Since there are more streamlines at the back (windward side)

of the sphere compared with that in the front (leeward), this results ..." (see line 64 of the revised manuscript).

2. Lines 63-64: "In addition to a backward stress...". It is unclear what backward stress means.

Reply: Thanks the reviewer for pointing out, we have changed "backward stress" to "backward force" (see line 69 of the revised manuscript).

3. Line 72: Symbol of diameter of sphere: I suggest replacing " σ " with "D".

Reply: Following the suggestion of the reviewer, the symbol " σ " for the sphere diameter has been replaced with "D" throughout the manuscript.

4. Line 84: Please define H and ω_H clearly here. It is hard to understand the difference between ω and ω_H .

Reply: We thank the reviewer for the suggestion. We have now added the following definitions of H and ω_H , which appears at Line 90 of the revised manuscript. "here $H = |\mathbf{H}|$ is the magnetic field strength, and ω_H is the angular velocity of H."

5. Figure 1d: To assist the reader, please add a dashed horizontal line (e.g., in red) corresponding to $k=0$.

Reply: We appreciate the reviewer's helpful suggestion. In the revised manuscript, a red dashed horizontal line corresponding to $k = 0$ has been added in Figure 1d to guide the reader's interpretation.

Reviewer #3 (Remarks to the Author):

The manuscript entitled "Observation of a backward sliding for rollers on surfaces in viscoelastic fluid" reports a counterintuitive phenomenon: rotating spheres ("rollers") translating backwards while rolling along a surface in viscoelastic fluids, an effect absent in Newtonian liquids. The authors combine experiments at different scales and with various viscoelastic media with numerical simulations based on the Giesekus model. They further show that viscoelastic normal stresses can generate an effective attraction to nearby surfaces, enabling the transmission of motion to larger passive spheres in a microscale gearing configuration. The topic falls well within the scope of Nature Communications, the results are interesting and novel, and the work adds to the literature on microrheology and active matter. However, several important issues must be addressed in a comprehensive revision before the manuscript can be considered further.

Reply: We are very thankful to the reviewer for the positive assessment of our manuscript. Below we answer his/her comments and suggestions point by point.

Comments

Clarity of Figure 1a (experimental schematic).

Figure 1a is too small and visually confusing for a reader encountering the problem for the first time. Because it introduces the setup and defines the physical problem, it should be enlarged and clearly annotated. At first reading, it is not clear whether the roller is fully immersed in the viscoelastic fluid or partly at a free surface; only later in the text does it become apparent that the particles sediment and roll along the bottom wall, fully immersed. This point should be explicitly stated early in the manuscript, and the schematic should include clear labels showing the wall, fluid region, coordinate system, rotation direction, and magnetic field orientation.

Reply: We are grateful for the reviewer's detailed and constructive suggestion. To make Figure 1a more self-explanatory, we have made the following revisions: (1) Enlarged Figure 1a by a factor of 2. (2) Changed the illustration of the fluid region from 2D to 3D with coordination systems. (3) Added a clear substrate (i.e. wall) at the bottom of the fluid region. (4) Clearly labeled the rotation direction of the roller and the magnetic field.

In addition to the changes made to the Figure 1a, we also added the following text in the figure caption to help the reader to understand our setup. "The roller sediments at the bottom surface of a thin cuvette with inner space 20 mm × 10 mm × 0.2 mm and filled with viscoelastic fluid. The rotating magnetic field $\mathbf{H}(t)$ caused the sphere to roll on the surface."

Giesekus model parameters and physical consistency.

The parameters used in the Giesekus model require much stronger justification. The mobility parameter $\alpha = 0.2$ is very large and inconsistent with the dilute PAAM solutions studied here, where α values below 0.01 are more realistic. Such a high α imposes strong shear-thinning and large second normal stress differences that are not representative of the experiment, even though this effect is partially suppressed by choosing $\beta = 0.9$. This discrepancy is evident in Fig. 3d, where the simulated coefficient k is underestimated by about one order of magnitude and the transition from positive to negative k occurs at Weissenberg numbers roughly three orders of magnitude smaller (≈ 10 in the simulation versus ≈ 1000 in the experiment). The parameter choice therefore does not capture the experimental regime. Moreover, for smaller and more realistic α values (e.g., 0.01 or less), the predicted normal stresses increase, which would shift the onset of backward sliding to lower Wi , not higher. If backward motion indeed appears at lower Wi in that case—as expected from the Oldroyd-B limit—the normal-stress-driven mechanism proposed here may not be correct. The authors should clarify whether backward sliding occurs for $\alpha = 0.01$ or in the Oldroyd-B model, and if so, whether the transition shifts to lower or higher Wi . This is critical for validating

the proposed physical mechanism.

Reply: We agree with the reviewer that, the mobility parameter $\alpha = 0.2$ is large for our micrometer roller experiments, where we have used a dilute PAAM solution ($c < 0.3$ g/L) with little shear thinning. However, this value (i.e. $\alpha = 0.2$) should be reasonable for our millimeter roller experiments (i.e. Fig. 3) where we have used relatively dense PAAM solution ($c = 1\sim 10$ g/L) which shows clear shear thinning. See the newly added supplementary Fig. 4 in our revised manuscript for the rheology measurements of our PAAM solutions.

Due to the numerical instability of the Oldroyd-B model at high Weissenberg numbers [1], our simulations are restricted to the Giesekus model with relatively large values of the mobility parameter α . This choice partially contributes to the quantitative differences between our simulation and experimental results. However, α has only a minor influence on the onset of backward motion. To demonstrate this, we performed additional simulations with $\alpha = 0.1$ and $\alpha = 0.01$ for rollers at a relatively large $h=1.4r$ (i.e. a relatively large gap of $0.4r$ between the roller and the ground, to avoid numerical instability). The corresponding $k - Wi$ relation is shown in the figure below. The data for $\alpha = 0.2$ corresponds to the results presented in Fig. 3d of our manuscript. As shown in the figure, the change of α from 0.2 to 0.01 has slightly increased the Wi_c , and the transition from positive k to negative k becomes a bit sharper. This is the right trend when we need to compare the results with our experiment. But the effect is relatively small.

The fact that α has only a minor influence can also be qualitatively explained by considering only the steady shear flow in the narrow gap between the roller and the wall, which dominates the hydrodynamic force acting on the roller. Under simple shear, both the viscous stress $\Gamma_s \sim \eta \dot{\gamma}$ and the first normal stress difference $N_1 \sim 2\tau \eta_p \dot{\gamma}^2$ remain finite as $\alpha \rightarrow 0$ [2]. Here η is the total viscosity, η_p is the polymer contribution to the viscosity, τ is the polymer relaxation time, and $\dot{\gamma}$ is the shear rate. We note that in flows dominated by extension, the total stress scales as $\Gamma \sim 2\eta_p \epsilon / \alpha$ [2], where ϵ is the extensional rate. This implies that the total stress increases monotonically as α decreases—consistent with the reviewer's remark. However, since the dynamics in our problem are governed primarily by shear rather than extension, this strong α -dependence does not arise.

Fig. 1. Dependence of k on Wi for the Giesekus model of different α , with $h = 1.4r$.

The reason that the simulated coefficient k is underestimated by one order of magnitude compared with that in experiments, is due to the fact that we have used a much larger value of h (i.e. the distance from the center of the roller to the ground) in simulation compared with that in experiments. For experiments shown in Fig. 3, the value of $h < 1.012r$ (i.e. the gap between the roller and the ground $< 0.012r$, see our newly added Supplementary Figure 3 and Video 6). While in simulation, all $h > 1.1r$ (i.e. the gap between the roller and the ground $> 0.1r$). The reason to use a much larger value of h in simulation is (also) to avoid numerical instability (Instability becomes very frequent when $h < 1.1r$, this becomes worse when $Wi > 1$). This larger value of h in simulation also leads to a much smaller value of the required critical Weissenberg number $Wi_c = 1 \sim 10$ in simulation compared with $Wi_c = 100 \sim 1000$ in experiments (consider that the friction f between the roller and the ground will significantly increase as h decrease, therefore it requires a much larger value of T_{\parallel} and hence a much larger value of Wi_c to achieve backward sliding in experiments). In fact, we do not mean to achieve a quantitative match between our experimental results in Fig. 1c and the simulation results in Fig. 1d. Our motivation to include Fig. 1d is to emphasize the qualitative match, i.e. both the millimeter-roller experiments and simulation reveal a transition from a positive k (forward motion) to a negative k (backward motion) at

certain Wi_c . This is already stated in our manuscript "Even though the numerical results have a much smaller k_0 due to the large sphere-to-surface gap (see Supplementary Fig. 3 and Video 6), they capture the essence of Eq. (3) which suggests that k would approach a negative k_∞ at large Wi ." at line 279 of the revised manuscript.

We thank the reviewer for mentioning the influence of the mobility parameter α , we have added the above figure and the relevant discussion in the Supplementary Information of the revised manuscript.

[1] Keunings, R., 1986. On the high Weissenberg number problem. *Journal of Non-Newtonian Fluid Mechanics*, 20, pp.209-226.

[2] Bird, R.B., Armstrong, R.C. and Hassager, O., 1987. *Dynamics of polymeric liquids*. Vol. 1: Fluid mechanics. 2nd edition, page 368.

Unclear geometry in Figure 2.

The geometry depicted in Figure 2 is difficult to understand. It is not clear what the "expansion" represents, what the domain boundaries are, or how the illustrated flow corresponds to the experimental setup in Figure 1a. The figure and caption should explicitly indicate whether the sphere is rotating above a flat wall or near a "cliff," what the boundary conditions are on each side, and what quantity is being visualized (velocity field, polymer stretch, or stress). A short paragraph in the text explaining the computational geometry and its relation to the experiments would greatly improve readability and credibility of the simulation results.

Reply: For the computational geometry of Figure 2a (i.e. the rolling of a sphere on a flat surface within viscoelastic fluid), the liquid domain reaches 60 sphere diameters away from the center of the roller in all directions (i.e. top, front, back, left, right) except to the surface below which is only $h = 1.15r$. The boundary condition at the sphere surface is set to be a rotating-sphere solid wall. Consider that in experiments the roller is moving backward (towards negative x direction) with velocity v , in simulation we use a moving frame where the entire liquid domain moves at a velocity $-v$ along the x direction relative to the fixed rotating sphere. This means we have inlet boundary at negative x direction and outlet boundary at positive x direction with velocity $-v$. The bottom wall is also moving with velocity $-v$ such that it is a no-slip boundary. All other boundaries are far-field boundary with zero pressure and zero velocity gradient.

For the computational geometry of Figure 2c (i.e. the rolling of a sphere on a flat surface towards a cliff inside viscoelastic fluid), the computational geometry is the same to that above except that here we choose $v = 0$ and a stair-like surface at the bottom with a single step that represents the cliff. The step height is 2 sphere diameters.

We thank the reviewer for point out the lack of clarity in the computational geometry. Following the suggestion of the reviewer, we have now added the relevant information (i.e. whether the sphere is rotating above a flat wall or near a "cliff," what the boundary conditions are on each side, and what quantity is being visualized) in the figure caption

to enhance the readability of our manuscript.

Review of: Observation of a backward sliding for rollers on surfaces in viscoelastic fluid

The paper investigates the backward sliding motion of magnetic rollers near surfaces in complex fluids. The authors present convincing experiments of the observed dynamics, as well as evidence of hydrodynamic coupling between the rollers and the surface. Numerical simulations complement the observations and provide some explanation for the dynamics, revealing tension force that opposes the natural friction force between the rollers and the surface. After carrying out a force analysis, the authors determined conditions under which the backward sliding motion is likely to occur.

Overall the paper is very well written, the experimental methodology leads to clear and strong evidence of the dynamics, and the theoretical analysis is sound. The results are noteworthy; to my knowledge, reports of backward motions are only now beginning to emerge in the low-Reynolds propulsion literature. [For instance, Della-Giustina *et al.* (**Physics of Fluids**, 2023) in the case of an active particle encapsulated in a droplet in a porous medium.] Although using a much different physical set-up compared to previous studies, the authors' findings provide further evidence of counterintuitive dynamical processes taking place in low-Reynolds number flows of complex fluids.

It is my opinion that the findings will expand the scientific understanding of microorganisms' dynamics in biologically-relevant environments. From a practical viewpoint, they also have the potential to guide the design of more efficient drug delivery cargos, thereby serving as the foundation for future studies. Audiences in fluid dynamics and related interdisciplinary fields will benefit from the paper's findings. I highly recommend the manuscript for publication in **Nature Communications**, and only have the following very minor comments.

1. The title could benefit from a bit more clarity. I would suggest "Observation of a backward sliding *motion* for rollers on surfaces in viscoelastic fluids".
2. Line 167: I suggest replacing the text "Eq. (2) fits nicely to" with "Eq. (2) agrees well with".
3. The authors note that Eq. (2) may not be valid at large polymer concentration. Is there a way to quantify the range of polymer concentration for which Eq. (2) remains a good approximation? In other words, what is considered large? (I presume the answer will depend on the size of the rollers.) A more detailed note on this limitation may be useful.